# SERPINA3-ANKRD11-HDAC3 pathway induced aromatase inhibitor resistance in breast cancer can be reversed by HDAC3 inhibition

Jing Zhou [1,4], Mengdi Zhu[1,4], Qi Wang [1], Yiyuan Deng[2], Nianqiu Liu [3], Yujie Liu[1] & Qiang Liu [1✉]

Endocrine resistance is a major challenge for breast cancer therapy. To identify the genes pivotal for endocrine-resistance progression, we screened five datasets and found 7 commonly dysregulated genes in endocrine-resistant breast cancer cells. Here we show that downregulation of serine protease inhibitor clade A member 3 (*SERPINA3*) which is a direct target gene of estrogen receptor α contributes to aromatase inhibitor resistance. Ankyrin repeat domain containing 11 (ANKRD11) works as a downstream effector of SERPINA3 in mediating endocrine-resistance. It induces aromatase inhibitor insensitivity by interacting with histone deacetylase 3 (HDAC3) and upregulating its activity. Our study suggests that aromatase inhibitor therapy downregulates SERPINA3 and leads to the ensuing upregulation of ANKRD11, which in turn promotes aromatase inhibitor resistance via binding to and activating HDAC3. HDAC3 inhibition may reverse the aromatase inhibitor resistance in ER-positive breast cancer with decreased SERPINA3 and increased ANKRD11 expression.

[1] Breast Tumor Center, Sun Yat-Sen Memorial Hospital, Sun Yat-Sen University, Yanjiang West Road 107#, 510120 Guangzhou, China. [2] The China-Japan Union Hospital of Ji Lin University, Changchun, China. [3] Kunming Medical University, Kunming, Yunnan, China. [4]These authors contributed equally: Jing Zhou, Mengdi Zhu. ✉email: liuq77@mail.sysu.edu.cn

Breast cancer has surpassed lung cancer as the most common cancer and leading cause of cancer-related deaths in women worldwide[1]. Estrogen receptor positive (ER+) breast cancer, whose proliferation is stimulated by 17β-estradiol (E2) through ER genomic and non-genomic action, takes up nearly 80% of all breast cancers. The most common pharmacological therapeutic strategies for patients with ER+ breast cancer are endocrine therapies including selective estrogen receptors α (ERα) modulators, such as tamoxifen, selective ERα down-regulators and aromatase inhibitors (AIs)[2]. For postmenopausal women, AIs including exemestane, anastrozole and letrozole, are currently the most commonly prescribed drugs in adjuvant hormonal therapy[3]. Despite the substantial clinical benefit of ERα-targeting therapy, intrinsic and acquired resistance represents a major challenge in ER+ breast cancer. Almost all metastatic ER+ breast cancer will sooner or later develop endocrine resistance[4,5]. Hence, it is of great clinical significance to better understand the molecular mechanisms driving endocrine resistance and develop strategies to effectively overcome it.

The zinc-dependent mammalian histone deacetylase (HDAC) is a family of epigenetic modulators comprising 18 enzymes that are active in removing acetyl groups from histones, inducing the formation of a compacted and transcriptionally repressed chromatin structure. HDACs can be classified into four groups based on sequence homologies. Class I includes HDAC1, 2, 3 and 8; class II includes HDAC4, 5, 6, 7, 9, 10; class III consist of seven sirtuins which possess a catalytic site structurally different from the other three classes; class IV includes only HDAC11[6,7]. Given the strong regulation of HDACs on gene expression, HDAC inhibitors (HDACis) showed therapeutic potential in numerous diseases, including cancers, myelodysplastic syndromes, neurological diseases and immune disorders[8].

Epigenetic remodeling is one of the key mechanisms underlying endocrine resistance[9]. In recent years, mounting evidence suggested high efficiency of HDACis in reversing anti-estrogen resistance in ER+ breast cancer. Several pan-HDACis were shown to possess more pronounced anti-proliferative effect in ER+ breast cancer than ER negative (ER−) subtype[10–13]. Later, Class I HDAC inhibitor FK228 was found to inhibit the growth of tamoxifen-resistant xenografts[14]. Pan-HDAC inhibitor vorinostat and panobinostat were shown to potentiate tamoxifen effect in ER+ breast cancer[15,16]. Furthermore, class I HDAC inhibitor entinostat was shown to sensitize breast cancer cells to AI, by upregulating ER, aromatase and downregulating HER-2[17,18]. Recently, more activated HDAC9 and HDAC3 were respectively shown to be associated with endocrine resistance[19,20]. It was also reported that HDAC3 could form a complex with ERα and inhibit apoptosis in MCF-7 cells[21].

Based on the extensive pre-clinical evidence, several clinical trials were performed to explore the feasibility of combing HDAC inhibitor and aromatase inhibitors in ER+ breast cancer. In 2019, a phase 3 ACE trial showed that HDAC1/2/3/10 inhibitor chidamide plus exemestane (an aromatase inhibitor) significantly improved progression-free survival of advanced ER+ breast cancer patients compared with exemestane monotherapy, highlighting the clinical efficiency of class I and II HDAC inhibitors in overcoming aromatase inhibitor resistance of breast cancer[22]. However, despite a significant 8.3-month survival benefit in the phase 2 trial[23], the addition of another HDAC 1/2 inhibitor entinostat to exemestane failed to provide a survival benefit to endocrine-resistant breast cancer patients in another phase 3 E2112 trial[24]. These contradictory results raised concerns over the clinical efficacy of HDACis in ER+ breast cancer treatment, and necessitate more detailed study of the target and mechanism of HDACs in reversing AI resistance.

To study the mechanisms of AI resistance, in vitro AI-resistant cell models were established by depriving breast cancer cell lines

of estrogen long term, which mimics the hormone withdrawal situation in breast cancer patients receiving AI-therapy[25–27]. By using the two long-term estrogen deprivation (LTED) derivatives generating from MCF-7 and T47D, we found that the downregulation of serine protease inhibitor serpin family A member 3 (SERPINA3), a member of the serine-protease inhibitor family, could induce endocrine-resistance through regulating HDAC3 activity in ER+ breast cancer. Whether SERPINA3 acts as a promoter or suppressor in tumor progression seems to be cancer-type-specific[28]. Positive correlation between SERPINA3 expression and patient poor prognosis was observed in numerous cancers, including melanoma[29], lung[30], colon[31], endometrial[32], gastric[33] and triple negative breast cancer[34]. On the other hand, SERPINA3-deficient cancer cells proliferated significantly in hepatocellular carcinoma[35]. The exact roles of SERPINA3 in different physiological or pathological scenarios are yet undetermined[28]. Further elucidation of SERPINA3-mediated cancer development and its involved pathway might shed light on the realm of cancer therapeutics.

In this study, *SERPINA3* was identified to be a direct target gene of ERα, and its downregulation enhanced the expression of ankyrin repeat domain containing 11 (ANKRD11), which resulted in elevated HDAC3 activity and AI resistance in ER+ breast cancer. This study highlights the significance of upregulated HDAC3 activity in AI resistance and could explain the conflicting results of several clinical trials using different HDACis in ER+ breast cancer. More importantly, our findings suggest that decreased SERPINA3 as well as increased ANKRD11 expression as biomarkers that may predict the efficiency of HDAC3 inhibitor in treating AI-resistant breast cancer.

## Results

**Transcriptome-wide screening of candidate genes associated with endocrine resistance in ER+ breast cancer.** The established LTED lines of MCF-7 and T47D, which are used to mimic AI-resistant cells, were examined for their endocrine-resistant properties by colony formation and cell counting assays. As shown in Fig. 1a, b, unlike parental MCF-7 and T47D cells whose growth could be inhibited by E2 deprivation, the LTED cell lines exhibited comparable proliferation rates in estrogen-free and complete medium. These results indicated the successful establishment of AI-resistant LTED cells. To identify potential genes responsible for anti-estrogen resistance, transcriptome analysis of MCF-7 and its tamoxifen-resistant (TAMR) and LTED lines using microarrays were performed. A total of 5,597 consistently dysregulated genes (fold change <0.8 or >1.25) were detected in MCF-7 TAMR and LTED lines[36] (GSE234546). These 5597 genes were further overlapped with the previously reported 366 differentially expressed genes between MCF-7/S0.5 and its 4 TAMR lines[37], and the 286 DEGs between patient-derived luminal breast cancer xenografts HBCx22 and its TAMR as well as ovariectomy resistant derivatives[38]. Finally, 9 common dysregulated genes were obtained (Table 1). The screening steps was shown in Fig. 1c. QPCR performed in MCF-7, T47D and their LTED lines confirmed the decreased expression of 7 genes, but not the upregulation of the other 2 genes, in LTED cells (Fig. 1d, e). Therefore, the 7 downregulated genes were selected for further investigation.

**SERPINA3 down-regulation confers AI resistance to ER+ breast cancer cells.** To determine the role of the 7 genes in AI resistance, we designed two siRNAs for each gene and transfected them into MCF-7 cell respectively. Cells were then cultured with or without E2 for 5 days. The inhibit efficiency of siRNAs were examined by QPCR at day 3 (Fig. 2a) and cell proliferation were

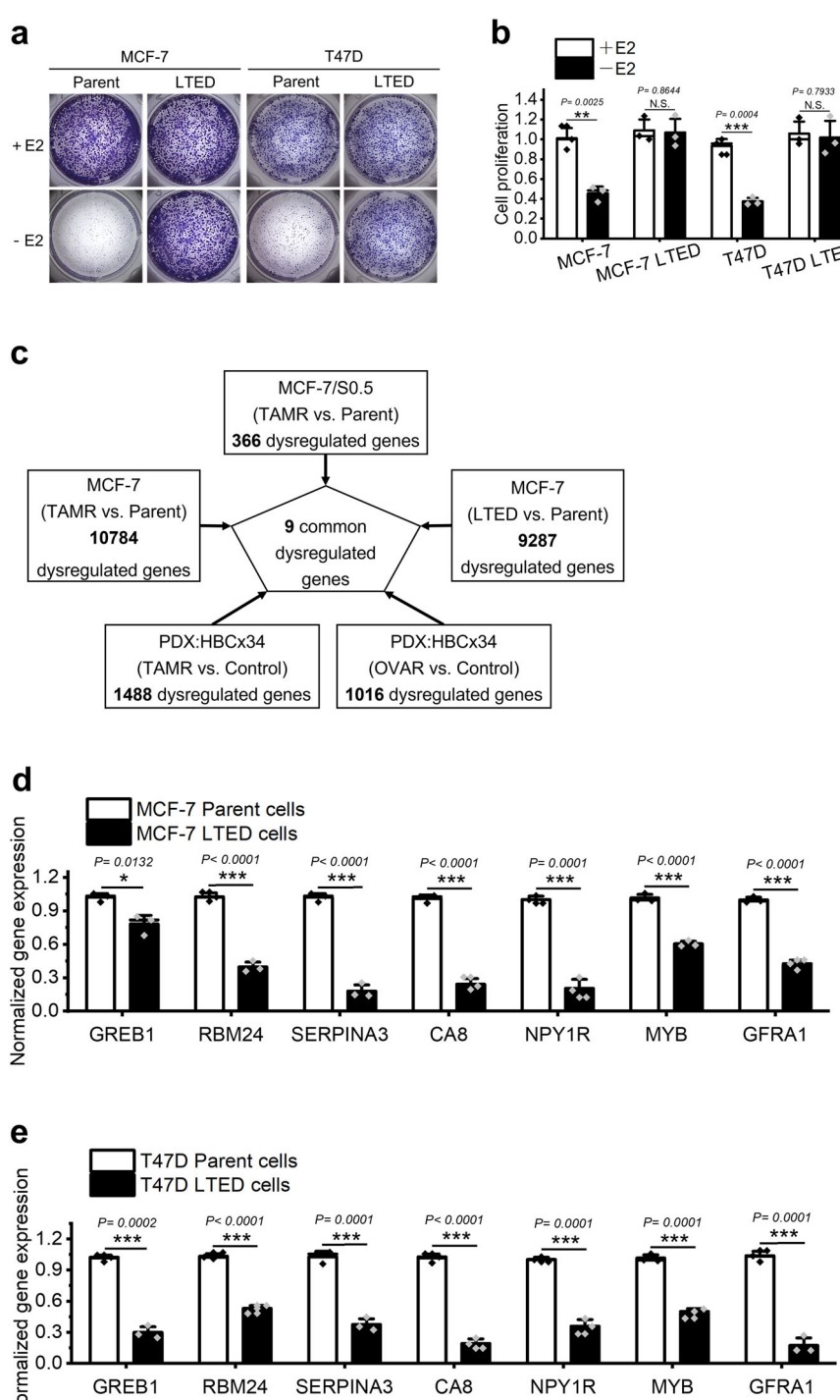

**Fig. 1 Screening of candidate genes responsible for endocrine resistance in human ER+ breast cancer. a** Estrogen deprivation for 14 days reduced colony formation of MCF-7 and T47D cells but not their LTED cells. **b** Estrogen deprivation for 5 days induced a clear reduction in cell proliferation of MCF-7 and T47D but not the LTED cells, as analyzed by cell counting assay. **c** Flowchart depicts how the 9 candidate genes were selected from various sets of RNA-seq or microarray data. **d, e** QPCR confirmed the uniform downregulation of the 7 genes in both MCF-7 and T47D LTED cells as compared with their parent cells. E2 estradiol, OVAR ovariectomy-resistance. Data are representative of $n = 3$ biologically independent experiments and presented as mean ± SD; statistical significance is determined by unpaired Student's $t$ test ($p < 0.05$*, $p < 0.01$**, $p < 0.001$***).

evaluated by cell counting at day 5. Results showed that *SERPINA3* is the only gene whose knock-down desensitized MCF-7 cells to E2 deprivation (Fig. 2b). Plate colony formation assay performed in MCF-7 and T47D cells confirmed that knock-down of SERPINA3 expression would result in E2-deprivation-resistance (Fig. 2c). In contrast, overexpression of SERPINA3 abolished anti-estrogen resistance, as shown by plate colony

formation assay conducted in MCF-7 and T47D LTED cells (Fig. 2d). Next, western blot verified decreased expression of SERPINA3 in two strains of LTED cells as compared with their parental lines (Fig. 2e). Microarray data of pre- and post-AI therapied breast tumors in GSE105777 and GSE153470 showed diminished expression of SERPINA3 in breast cancer patient samples after AI treatment (Fig. 2f, g)[39,40]. Furthermore,

**Table 1 The 9 genes commonly dysregulated in endocrine-resistant breast cancer cells from 5 sets of gene expression profiling data.**

| Official symbol | Official full name | Gene ID | Regulation |
|---|---|---|---|
| NPY1R | Neuropeptide Y receptor Y1 | 4886 | Down |
| GREB1 | Growth regulating estrogen receptor binding 1 | 9687 | Down |
| MYB | MYB proto-oncogene, transcription factor | 4602 | Down |
| CA8 | Carbonic anhydrase 8 | 767 | Down |
| SERPINA3 | Serpin family A member 3 | 12 | Down |
| RBM24 | RNA binding motif protein 24 | 221662 | Down |
| GFRA1 | GDNF family receptor alpha 1 | 2674 | Down |
| DUSP10 | Dual specificity phosphatase 10 | 11221 | Up |
| TFAP2A | Transcription factor AP-2 alpha | 7020 | Up |

according to the data retrieved from TCGA database (Supplementary Data 1), lower expression of SERPINA3 is specifically correlated with poorer prognosis in both luminal A and luminal B subtypes of ER+ breast cancer (Fig. 2h, i), but not ER- subtype (Supplementary Fig. 1). All above results together suggest that the expression of SERPINA3 is negatively associated with endocrine resistance in ER+ breast cancer. In conclusion, these data jointly suggest the contribution of decreased SERPINA3 to AI-resistance in ER+ breast cancer.

**SERPINA3 expression is directly regulated by ERα.** Given that endocrine therapy targets ERα pathway and SERPINA3 expression is closely related with endocrine therapy, we asked if SERPINA3 expression is regulated by ERα. According to data retrieved from cBioPortal, the expression of ESR1, which encoded ERα, and SERPINA3 were positively related in breast cancer, as evidenced by a Pearson correlation coefficient of 0.47 (Fig. 3a). Moreover, SERPINA3 expression was significantly higher in ER+ than ER− breast cancer (Fig. 3b), which also demonstrated a positive correlation between ER pathway and SERPINA3 expression. In addition, the expression of SERPINA3 was downregulated in LTED cells (Figs. 1d and 2e) in which ERα activity was inhibited due to the decreased ERα expression (Fig. 3c, d) and estrogen deprivation, implying an inhibition effect of ERα pathway on SERPINA3 expression. In line with this, siRNA inhibition of ERα downregulated SERPINA3 expression at both mRNA and protein levels in MCF-7 and T47D cells (Fig. 3e–g). Moreover, SERPINA3 mRNA expression gradually decreased with the extension of estrogen-deprived incubation (Fig. 3h). These findings collectively indicate that the expression of SERPINA3 is regulated by ERα pathway.

To further determine if SERPINA3 expression is directly regulated by transcription factor ERα, we used JASPAR software to check if there is any possible ERα binding site on SERPINA3 promoter. It was predicted that the fragment located at −462 bp to −446 bp upstream of SERPINA3 transcription start site, which shares high sequence similarity with ERα-specific binding motif, as the most likely ERα binding site (Fig. 3i, j). Luciferase reporter analysis was subsequently used to confirm the interaction between ERα and its predicted binding region on SERPINA3 promoter. Results showed that ERα overexpression induced a ~6-fold increase in luciferase activity over control in 293T cells under E2 treatment, moreover, this increase was abated by site-directed mutation of −462 bp to −446 bp upstream of SERPINA3 transcription start site (Fig. 3k). Collectively, these results suggest

that ERα regulates and enhances expression of SERPINA3 by directly binding to its promoter region at −462 bp to −446 bp upstream of its transcription start point.

**ANKRD11 mediates SERPINA3 down-regulation induced AI-resistance.** To determine the molecules working downstream of SERPINA3 in inducing AI resistance, we first identified 61 genes whose expressions are negatively correlated with SERPINA3 in breast cancer by using the cBioPortal database. Among them, 29 genes' high expression indicate poor prognosis in ER+ breast cancer, according to bc-GenExMiner v4.9 database. Subsequent QPCR further filtered out 5 genes which are consistently upregulated in MCF-7 and T47D LTED cells, from the 29 ones, to be candidate downstream effectors for further study (Table 2). The whole screening procedure was shown in Fig. 4a.

QPCR experiments performed in MCF-7 and T47D revealed that expressions of ANKRD11 and dysferlin (DYSF) were negatively regulated by SERPINA3 (Fig. 4b, c). Subsequent loss-of-function studies using siRNA in MCF-7 LTED and T47D LTED cells showed that ANKRD11, but not DYSF, contributed to E2-deprivation resistance (Fig. 4d, e). Simultaneous knock-down of ANKRD11 with SERPINA3 significantly abolished SERPINA3 inhibition induced E2-deprivation resistance in MCF-7 and T47D cells (Fig. 4f, g). Western blot showed increased ANKRD11 expression in SERPINA3-knockdown MCF-7 and T47D parental cells, confirming the negative regulation of ANKRD11 by SERPINA3 (Fig. 4h). Likewise, ANKRD11 expression was upregulated in LTED cells than their parental cells, according to QPCR and western blot analysis (Fig. 4i, j). These data strongly support that SERPINA3 inhibition induced ANKRD11 upregulation involved in endocrine-resistance development in ER+ breast cancer.

As for the clinical data support, a negative correlation between the expressions of ANKRD11 and SERPINA3 in breast cancer was shown in Fig. 4k. Differential expression of ANKRD11 in ER+ and ER− breast cancer was shown in Fig. 4l. The data from GSE147271 showed a downregulation of SERPINA3 and upregulation of ANKRD11 after endocrine treatment (Fig. 4m, n)[41], which is in line with our conclusion that ANKRD11 expression is negatively regulated by ERα-SERPINA3 axis. Furthermore, higher expression of ANKRD11 was associated with poor prognosis in luminal A subtype of ER+ breast cancer (Fig. 4o) but opposite outcome in ER− subset (Supplementary Fig. 2a), implying a role of ANKRD11 in mediating resistance to endocrine therapies in ER+ breast cancer.

In conclusion, the bioinformatic and experimental analyses suggest that ANKRD11 works downstream of ERα-SERPINA3 pathway to promote estrogen-deprivation-resistance (AI resistance) in ER+ breast cancer.

**ANKRD11 mediates AI resistance by promoting HDAC3 activity.** It had been reported that ANKRD11 promoted histone deacetylase activity of HDAC3 in neural precursors[42]. Thus, a series of experiments was designed to investigate if ANKRD11-HDAC3 axis existed and induced AI resistance in ER+ breast cancer. Firstly, western blot showed a negative correlation between acetylation of H3K9, which is exclusively targeted by HDAC3[43], and AI sensitivity in MCF-7, T47D and their LTED lines (Fig. 5a). Secondly, inhibiting ANKRD11 expression enhanced H3K9 acetylation without changing histone H3 and HDAC3 expression in two LTED cells (Fig. 5b). Thirdly, immunoprecipitation results showed that ANKRD11 directly bound to HDAC3 (Fig. 5c). These results suggest that ANKRD11 enhanced HDAC3 activity by physically interacting with it, without changing its expression level. Furthermore, the role of

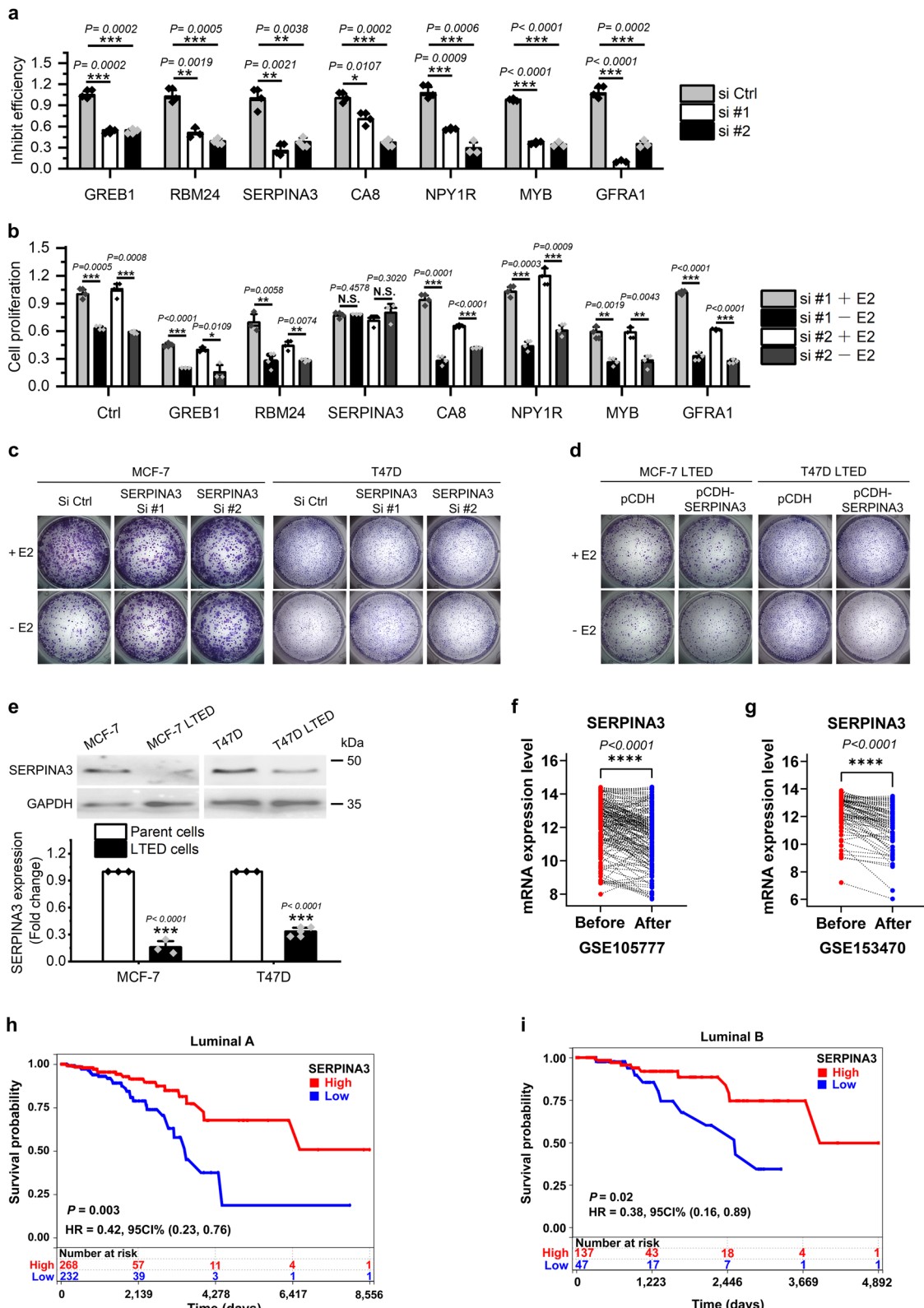

HDAC3 in AI resistance was determined in MCF-7, T47D and their LTED cells with a HDAC3 selective inhibitor RGFP966 or HDAC3 siRNAs. It was found that RGFP966 showed significant anti-proliferation effect on two LTED cells under estrogen deprivation, and the LTED cells were much more sensitive to RGFP966 than their parent cells (Fig. 5d, e). Similar results were observed with HDAC3 siRNAs (Fig. 5f). The results indicate a positive role of HDAC3 in enhancing AI resistance of ER+ breast cancer. Moreover, SERPINA3 knockdown-induced estrogen insensitivity could be abrogated by RGFP966 treatment under estrogen deprivation in MCF-7 cells (Fig. 5g), as well in T47D cells (Fig. 5h), further confirming the role of HDAC3 as the downstream effector in SERPINA3-ANKRD11 mediated AI-resistance.

**Fig. 2 SERPINA3 downregulation promotes AI-resistance in human ER+ breast cancer. a** The inhibit efficiencies of the siRNAs were detected by QPCR in MCF-7 cells. **b** MCF-7 cells were transfected with the siRNAs respectively and deprived of estrogen for 5 days. Cell proliferation was determined by counting. **c** T47D and MCF-7 cells were transfected with *SERPINA3*-targeting siRNAs and deprived of E2 or not for 7–10 days before crystal violet staining. **d** MCF-7 LTED and T47D LTED cells were transfected with SERPINA3-expressing plasmid and deprived of E2 or not for 7–10 days before crystal violet staining. **e** Downregulation of SERPINA3 in AI-resistant LTED lines were confirmed by western blot. According to the publicly available expression profiling data of clinical samples (**f**) GSE105777 (*n* = 172) and (**g**) GSE153470 (*n* = 112), SERPINA3 was downregulated in ER+ breast cancer after 2-week presurgical AI therapy. According to the data of 1057 breast cancer patients retrieved from TCGA database, low SERPINA3 expression correlates with poor survival in patients with both (**h**) luminal A and (**i**) luminal B subtypes of ER+. Data are representative of *n* = 3 biologically independent experiments and presented as mean ± SD; statistical significance is determined by unpaired Student's *t* test ($p < 0.05$ *, $p < 0.01$ **, $p < 0.001$ ***); statistical significance of microarray gene expression data of paired samples from GSE153470, GSE105777 are determined by paired Student's *t* test ($p < 0.05$ *, $p < 0.01$ **, $p < 0.001$ ***).

In summary, these findings demonstrate that AI therapy targeting ERα pathway down-regulates the expression of SERPINA3, which in turn up-regulates the expression of ANKRD11 and subsequently enhance the histone deacetylation activity of HDAC3, resulting in elevated histone deacetylation and further AI resistance development that can be reversed by HDAC3 inhibition (Fig. 6).

## Discussion
For ER+ breast cancer, the intrinsic or acquired endocrine resistance remains a major clinical challenge to be overcome, and the currently identified mechanisms are still far from sufficient to meet clinical needs. The development of an appropriate in-vitro cell model is key for effective investigation of the underlying molecular machineries. In clinical practice, AIs are typically administered in postmenopausal breast cancer patients to inhibit estrogen synthesis in the peripheral tissues such as adipose tissue, breasts and skin, thereby decrease blood estrogen concentrations and prevent cancer cell growth[3], but not directly act on the cancer cells themselves because of low or no expression of endogenous aromatase in cancer cells[25], which could be manifested by the insensitivity of ER+ breast cancer cells to therapeutic blood concentrations of the letrozole (50–100 ng/ml), anastrozole (10–40 ng/ml) and exemestane (10–20 ng/ml)[44–46] (Supplementary Fig. 3). LTED model was established by maintaining ER+ breast cancer cells in estrogen deprivation condition for over one year to mimic the physiological estrogen-depletion situation induced by AI therapy in patients. The validity of LTED model in mimicking AI-resistance has been widely accepted, as evidenced by the usage of this model in a number of high-quality AI resistance studies[25–27,47–49]. In another way of establishing AI-resistant model, specific AI was used to directly treat ER-positive breast cancer cells with ectopically overexpressed aromatase. This methodology is artificial and very different from factual physiological environment. The exclusive resistance to each specific AI of letrozole-, exemestane- and anastrozole-resistant MCF-7 cell lines[50] revealed the limitations of this model in mimicking patients' AI-resistance because patients who are resistant to letrozole are also resistant to anastrozole. Hence, in this study, the LTED model which more closely mimic the physical estrogen-depletion situation induced by AI therapy in patients was established and employed. By utilizing the two AI-resistant cell model MCF-7 LTED and T47D LTED, we revealed that down-regulated expression of *SERPINA3*, an ERα target gene, confers endocrine-resistance to breast cancer cells during AI therapy by enhancing ANKRD11 expression and HDAC3 deacetylase activity. Despite ample preclinical evidence supporting the remarkable efficiency of various pan-HDACis in overcoming endocrine resistance, two phase 3 clinical trials using different HDACis in endocrine-resistant breast cancer got conflict results, which necessitates further in-depth investigation on the molecular mechanism of HDACs in promoting endocrine-resistance for more safe and effective administration of HDACis in clinic. This study identified a

HDAC member, HDAC3, as a downstream target of ERα pathway that contributes to AI resistance in ER+ breast cancer, which may provide explanations for the contradictory results between clinical trials ACE and E2112. It also suggests HDAC3 as a specific and actionable target to reverse AI resistance in breast cancer. Whether effective HDAC3 inhibition is sufficient to provide additional benefits with AI therapy in endocrine-resistant breast cancer is a question worth further clinical assessment.

Although an old report described a stronger expression of HDAC3 in ER− than ER+ breast cancer[51], our QPCR data denied the regulatory effect of ERα pathway on HDAC3 expression (Supplementary Fig. 4). Data retrieved from bc-GenExMiner v4.9 even indicates a slight decrease of HDAC3 in ER− breast cancer compared with ER+ subtype (Supplementary Fig. 5). Online microarray data GSE105777 and GSE153470 showed that there were no changes in HDAC3 expression after AI therapy as well[39,40]. And in line with these results, ANKRD11 knockdown had no effect on HDAC3 expression (Fig. 5b), which confirmed that ANKRD11 facilitated HDAC3 activity by post-transcriptional mechanisms but not regulating its expression. These findings suggested that it is infeasible to use HDAC3 expression level as a prognostic indicator of HDAC3 inhibitors in breast cancer. However, ANKRD11 expression level may serve as a substitute marker. Notably, despite consistent upregulation of ANKRD11 in in-vitro established AI-resistant LTED cell lines, ANKRD11 was not necessarily upregulated in patients' tumors after hormone therapy, as manifest by its unchanged expression in datasets GSE105777/GSE153470 and upregulation in dataset GSE147271[39–41]. This inconsistency may be attributed to complicated physiological context in vivo and further necessitates detection of ANKRD11 expression before HDAC3-targeting therapy. In addition, according to TCGA data, high expression of ANKRD11 predicts worse prognosis in luminal A breast cancer but opposite outcome in luminal B and ER− subtypes which possess higher level of Ki67. The distinguished roles of ANKRD11 in Ki67-high and -low group suggest a strong disturbance of Ki67 related pathway to ANKRD11 function and highlight the necessity of restrictive administration of HDAC3 inhibitor in luminal A breast cancer.

While most studies demonstrated a positive role for SERPINA3 in tumor progression, our findings showed that SERPINA3-deficiency promotes AI resistance in ER+ breast cancer, suggesting a negative role of SERPINA3 in hormone therapy resistance induction. It seems that differential cellular and subcellular localization, namely secretory, cytosolic and nuclear, of SERPINA3 may contribute to its diverse functions. SERPINA3 is predominantly localized in cytoplasm in glioma, melanoma, lung and colon tumor tissues where its high expressions are associated with poor prognosis[28]. While in liver cancer, in which SERPINA3 relatively highly localized in nuclei, this protein indicates a better patient survival[52]. What is perhaps even more interesting is that, in MCF-7 cells, SERPINA3 was found to translocate to nuclei under estrogen deprivation[28]. The nuclear translocation of SERPINA3 during anti-

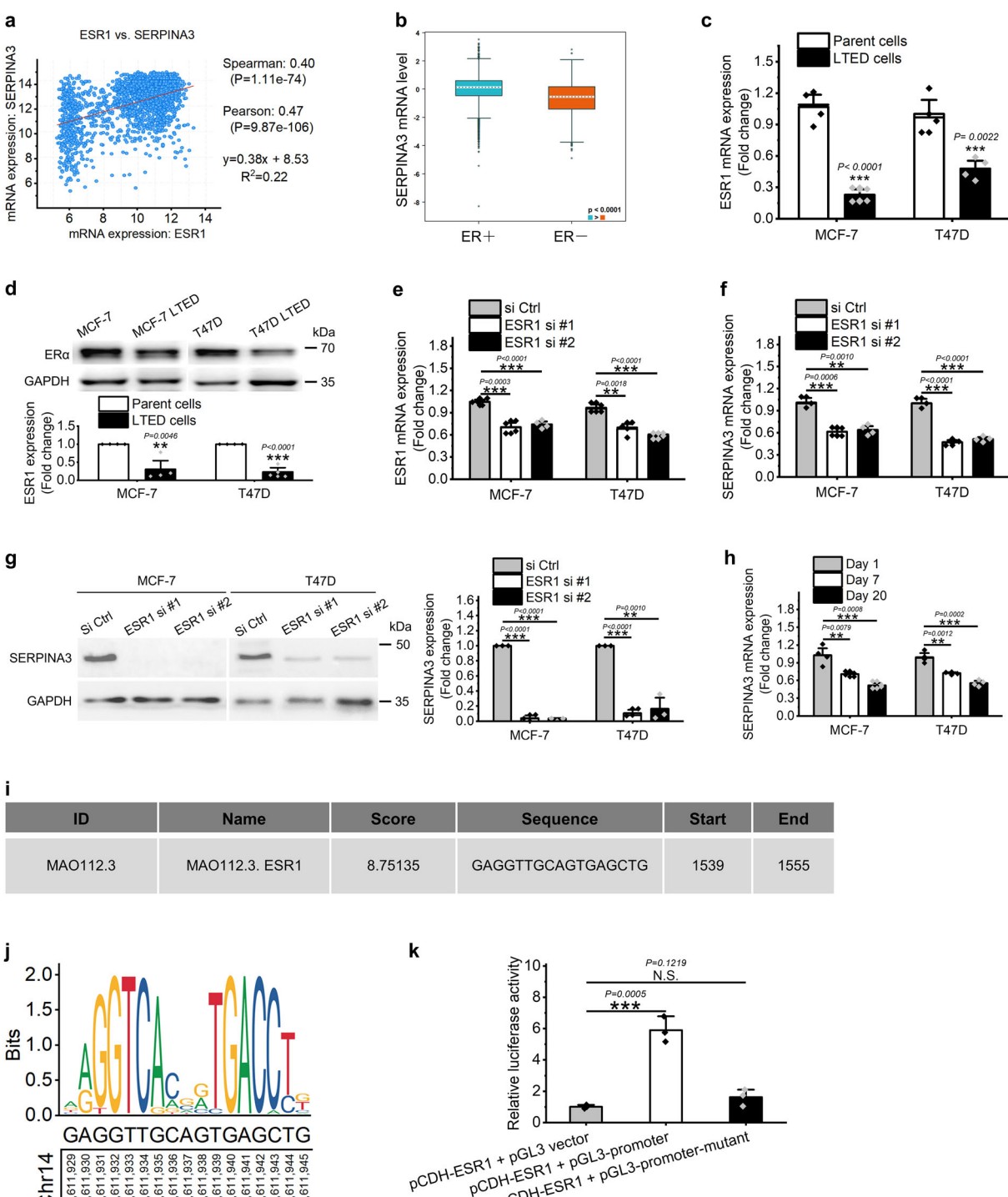

**Fig. 3 *SERPINA3* is an ERα target gene in human breast cancer. a** The expression of SERPINA3 is positively correlated with ESR1 in breast cancer (retrieved from cBioPortal). **b** SERPINA3 exhibits significant higher expression in ER+ breast cancer ($n = 7247$) than ER− subtype ($n = 2710$) (retrieved from bc-GenExMiner v4.9). **c** QPCR and **d** Western blot results revealed a decrease of ERα expression in LTED cells, when compared with their parental cells. **e** The expression of ESR1 could be effectively inhibited by the siRNAs in MCF-7 and T47D cells. **f** QPCR and **g** western blot results showed that siRNA knockdown of ESR1 in MCF-7 and T47D effectively downregulated SERPINA3 expression. GAPDH and SERPINA3 were blotted from different gels for their close molecular weight. **h** *SERPINA3* mRNA expression decreased with extended estrogen deprivation incubation in MCF-7 and T47D. **i** The details of the predicted ERα binding sequence in *SERPINA3* promoter obtained from JASPAR. **j** Comparison of ERα binding motif and the predicted ERα binding sequence in *SERPINA3* promoter. **k** Dual luciferase reporter assay verified the physical interaction of ERα and *SERPINA3* promoter at −462 to −446 bp. The promoter plasmid or its mutant type were co-transfected with pCDH empty vector or ESR1-expressing plasmid as indicated. Luciferase assays were then performed. pCDH-Vector: pCDH-CMV-MCS-EF1-puro; pCDH-ESR1: pCDH-CMV-MCS-EF1-puro-ESR1. Data are representative of $n = 3$ biologically independent experiments and presented as mean ± SD; statistical significance is determined by unpaired Student's $t$ test ($p < 0.05$ *, $p < 0.01$ **, $p < 0.001$ ***).

**Table 2 The screened 5 potential downstream effector of SERPINA3 in inducing endocrine resistance in human breast cancer.**

| Gene symbol | Official full name | Gene ID | Regulation in MCF-7 TAMR | Regulation in MCF-7 LTED | Regulation in clinical samples |
|---|---|---|---|---|---|
| E2F4 | E2F transcription factor 4 | 1874 | Up | Up | Up |
| KLF11 | KLF transcription factor 11 | 8462 | Up | Up | Up |
| NPL | N-acetylneuraminate pyruvate lyase | 80896 | Up | Up | Up |
| ANKRD11 | Ankyrin repeat domain containing 11 | 29123 | Up | Up | Up |
| DYSF | Dysferlin | 8291 | Up | Up | Up |

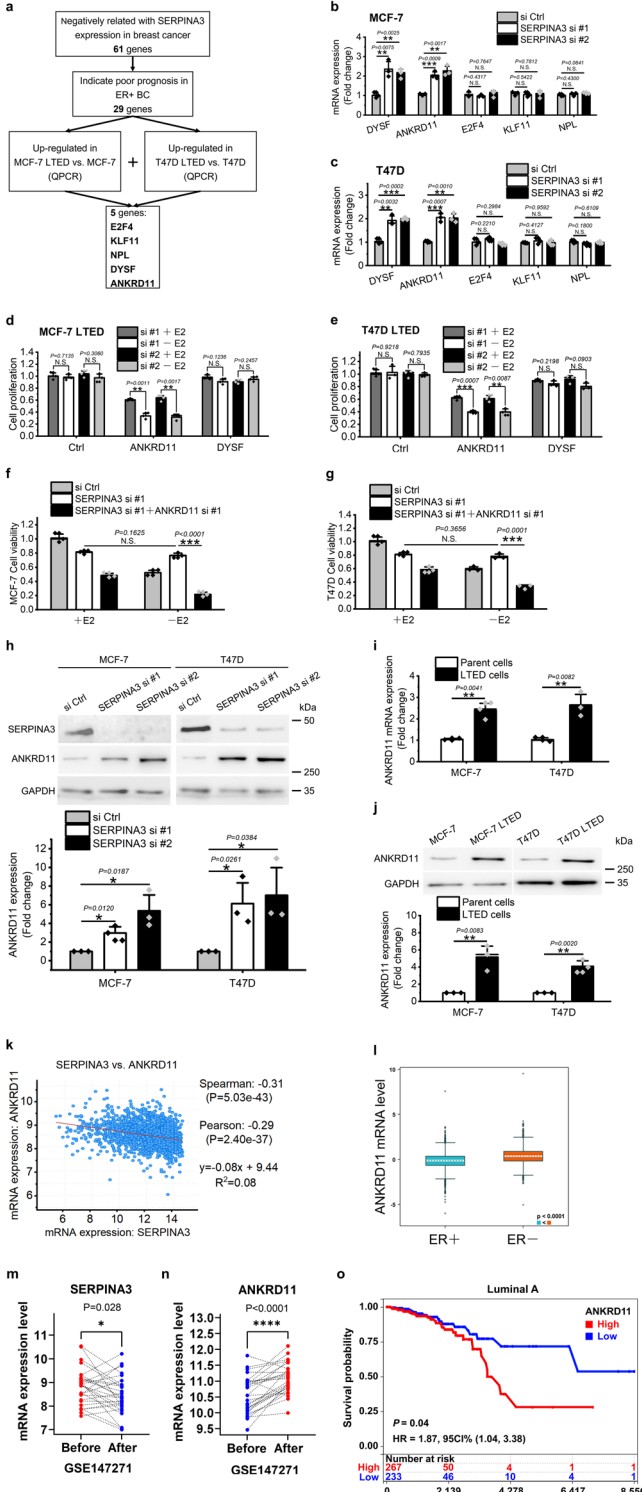

**Fig. 4 SERPINA3 downregulation drives AI resistance by ANKRD11 upregulation. a** Flowchart depicting work steps to identify SERPINA3 downstream effectors in inducing AI-resistance. *SERPINA3* correlated genes in breast cancer were retrieved from cBioPortal. The genes indicating poor prognosis in ER+ breast cancer were screened by using bc-GenExMiner v4.9. The expressions of the 5 genes were detected by QPCR after siRNA inhibition of SERPINA3 in (**b**) MCF-7 and (**c**) T47D. LTED cells of (**d**) MCF-7 and (**e**) T47D were transfected with the siRNAs of ANKRD11 and DYSF respectively and deprived of estrogen or not for 5 days, then cell proliferation was examined by cell counting. ANKRD11 inhibition restored sensitivity to estrogen deprivation in SERPINA3 knock-down (**f**) MCF-7 and (**g**) T47D cells. **h** Western blot showed that ANKRD11 was upregulated by SERPINA3 knockdown in MCF-7 and T47D. ANKRD11, SERPINA3 and GAPDH were blotted from 3 different gels. **i** QPCR and **j** western blot showed that ANKRD11 is overexpressed in LTED cells. ANKRD11 and GAPDH were blotted from different gels. **k** Expressions of SERPINA3 and ANKRD11 are negatively related in breast cancer (retrieved from cBioPortal). **l** ANKRD11 exhibits significant lower expression in ER+ than ER − subtype (retrieved from bc-GenExMiner v4.9). According to clinical samples' gene expression profiling data in GSE147271 ($n = 28$), the expression of (**m**) SERPINA3 was significantly downregulated and (**n**) ANKRD11 was significantly upregulated after presurgical tamoxifen therapy for 20.7 (±9.6) days. **o** High expression of ANKRD11 correlates with poor survival in patients with luminal A ER+ breast cancer according to TCGA database. Data are representative of $n = 3$ biologically independent experiments and presented as mean ± SD; statistical significance is determined by unpaired Student's *t* test ($p < 0.05$ *, $p < 0.01$ **, $p < 0.001$ ***); statistical significance of microarray gene expression data of paired samples from GSE147271 is determined by paired Student's *t* test ($p < 0.05$ *, $p < 0.01$ **, $p < 0.001$ ***).

estrogen therapy and the resulted benign prognostic effect make sense of the negative role of SERPINA3 in AI resistance development in ER+ breast cancer. However, more research is required to verify this deduction. This study for the first time identified *SERPINA3* as a target gene of ERα and uncovered a previously unappreciated function of SERPINA3 in endocrine resistance.

In line with the suggested proliferation-promoting effect of SERPINA3 in various cancers[28], a slight decrease in cell proliferation of the SERPINA3-knockdown MCF-7 cells was observed in this study. It seems that AI-treated cells improved survival or AI resistance with sacrificing partial proliferation ability by SERPINA3 down-regulation. But in fact, MCF-7 and T47D LTED lines did not exhibit a very significant decrease in proliferation as compared to parent lines. The PI3K-AKT-mTOR pathway, which has been shown to be upregulated in endocrine-resistant breast cancer cells[53], may act as a compensatory pro-proliferation pathway following SERPINA3 inhibition.

Numerous studies provided solid evidence for the promotional role of SERPINA3 in pathogenesis of neurodegenerative disorders[54]. Apart from SERPINA3, ANKRD11 also has been showed to facilitate neural proliferation and differentiation in neural precursors via HDAC3 activation[42]. And it is even

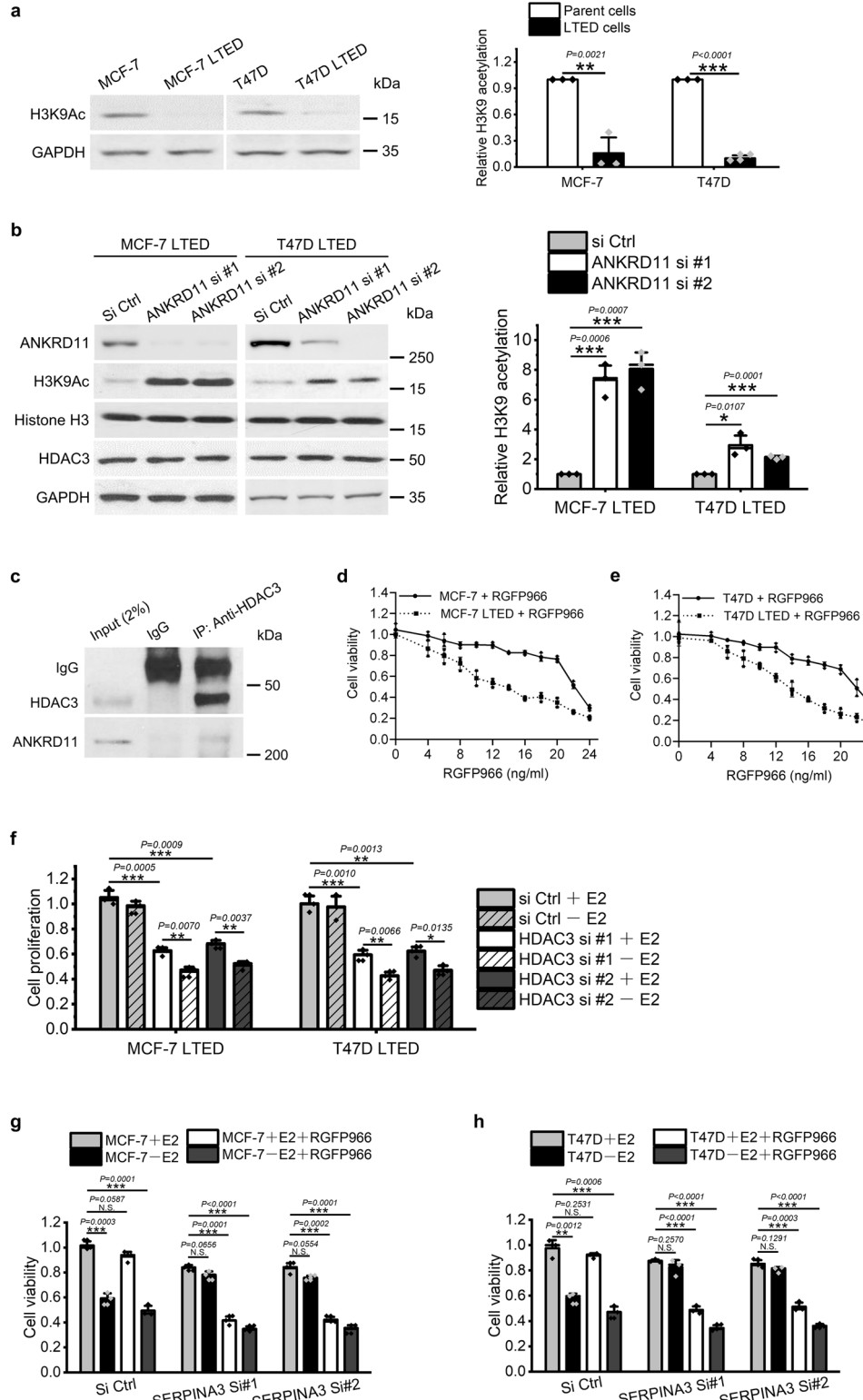

**Fig. 5 ANKRD11-enhanced HDAC3 activity endows AI-resistance to breast cancer. a** Western blot showed a decreased acetylation of H3K9, a HDAC3 specific target, in LTED cells. **b** Inhibiting ANKRD11 by siRNA significantly suppressed H3K9 acetylation without changing HDAC3 expression in LTED cells. H3K9Ac and GAPDH were separated and blotted from one gel, Histone H3 and HDAC3 were separated and blotted from another gel, and ANKRD11 were separated individually from one gel. **c** Immunoprecipitation confirmed the physical interaction between ANKRD11 and HDAC3 in MCF-7 LTED. IP immunoprecipitation, IgG immunoglobulin G. RGFP966 showed much higher anti-proliferation effect in (**d**) MCF-7 LTED and (**e**) T47D LTED cells than their parent cells. **f** More specific inhibition of HDAC3 using siRNA showed similar results as RGFP966. RGFP966 re-sensitized SERPINA3-knockdown- (**g**) MCF-7 and (**h**) T47D cells to estrogen-deprivation. Data are representative of $n = 3$ biologically independent experiments and presented as mean ± SD; statistical significance is determined by unpaired Student's $t$ test ($p < 0.05$ *, $p < 0.01$ **, $p < 0.001$ ***).

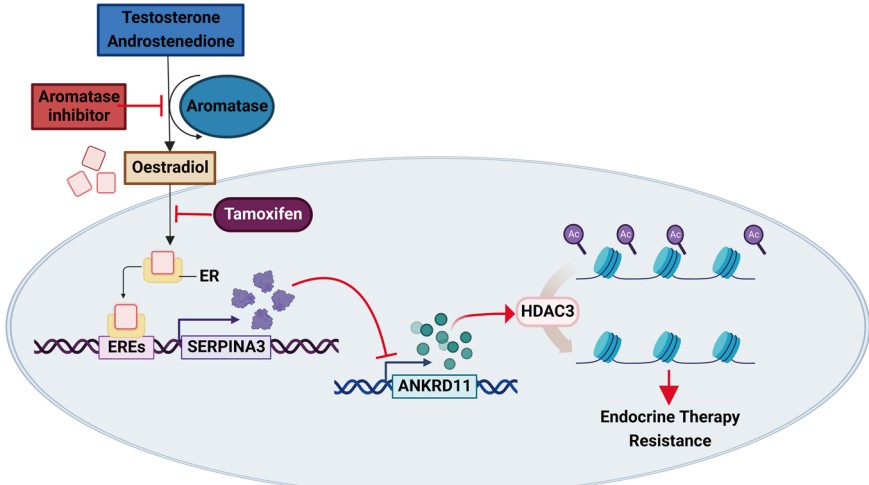

**Fig. 6 Schematic illustration of ERα-SERPINA3-ANKRD11-HDAC3 pathway.** Schematic showing the proposed mechanism of anti-estrogen-therapy-induced SERPINA3 expression loss driving endocrine resistance in human ER+ breast cancer.

| Table 3 The sequences of siRNAs used in this study. | | |
|---|---|---|
| **Target gene (official symbol)** | **siRNA** | **Target sequence** |
| GREB1 | siRNA 1 | GCCATTCGTGTGCTTCCAT |
| | siRNA 2 | CCTCCTACAAAGCAATATT |
| RBM24 | siRNA 1 | GCCTTTGGTGTTCAACAACTT |
| | siRNA 2 | TGGAGCTGCATACGCACAATA |
| SERPINA3 | siRNA 1 | CCCAAGATACTCATCAGTCAA |
| | siRNA 2 | GCATCACCTGACTATACCTTA |
| CA8 | siRNA 1 | CCAGTCTCCTATTAACCTAAA |
| | siRNA 2 | CGGGATTACTGGGTGTATGAA |
| NPY1R | siRNA 1 | CACUCUAAUUUCUCAGAGA |
| | siRNA 2 | GGCGATGTGTAAGTTGAAT |
| MYB | siRNA 1 | GGTCGAACAGGAAGGTTAT |
| | siRNA 2 | GTCCGAAACGTTGGTCTGT |
| GFRA1 | siRNA 1 | ATAATTACCATTGGAAATACAGAGG |
| | siRNA 2 | GCAAATTTACAGATCTCGCCT |
| ESR1 | siRNA 1 | CCGTAATGATTCTATAATG |
| | siRNA 2 | GCCTGGTGATTATTCATTT |
| ANKRD11 | siRNA 1 | CTGATAAGAATGTCATCTTCTCC |
| | siRNA 2 | CTGGTGAATTACATCTTCTTTAA |
| HDAC3 | siRNA1 | CCGCCAGCAAUCUUUGAATT |
| | siRNA2 | GCUUCACCAAGAGUCUUAATT |

| Table 4 The sequences of primers used for QPCR in this study. | | |
|---|---|---|
| **Gene (official symbol)** | **Primer** | **Sequence** |
| NPY1R | Forward | TACCAGCGGATCTTCCCCAC |
| | Reverse | AATTTGTCTTTTTCGCTCCTGC |
| GREB1 | Forward | GGGATCTTGTGAGTAGCACTGT |
| | Reverse | CATACGGGAAGGAGGTCACG |
| MYB | Forward | CACAGAACCACACATGCAGC |
| | Reverse | GCAGAGATGGGAGTGGAGTGG |
| CA8 | Forward | CCCAGCTACAGATAGAAGAATTCG |
| | Reverse | CACTAAGAGGCTGAGTGGGC |
| SERPINA3 | Forward | ATGGGAGATGCCCTTTGACC |
| | Reverse | CATGGGCACCATTACCCACT |
| RBM24 | Forward | AGATGCACACGACCCAGAAG |
| | Reverse | CGATCTCGCCGAAGACCTC |
| GFRA1 | Forward | CTGCGCATTTACTGGAGCAT |
| | Reverse | GAATGTGCTCCACTTGCTGAA |
| ESR1 | Forward | TGTTGAAACACAAGCGCCAG |
| | Reverse | GGTTGGCAGCTCTCATGTCT |
| ANKRD11 | Forward | TCCTTCGCACCTTCCAGTTC |
| | Reverse | GGTTCTTGGCCTTGTGCTTG |
| HDAC3 | Forward | AATGCCTTCAACGTAGGCGA |
| | Reverse | GGGTTGCTCCTTGCAGAGAT |

interesting to know that upregulated cytoplasmic HDAC3 is an independent prognostic factor for brain metastasis-free survival of breast cancer patients[55]. Being consistent with these studies, nervous system development pathway was upregulated in endocrine-resistant breast cancer cells in our GO enrichment analysis. These findings indicate the possibility of neural development or differentiation in endocrine-resistant breast cancer cells. Considering the non-negligible importance of neuroendocrine differentiation in anti-androgen therapy resistance in prostate cancer[56], it is then worthy to investigate whether and how neural differentiation happens in ER+ endocrine-resistant breast cancer and thereafter what its role is in endocrine-resistance and brain metastasis.

In summary, our study reveals the unacknowledged role of SERPINA3-ANKRD11-HDAC3 axis in AI-resistance which can be overcome by HDAC3 inhibition. This study provides the rationale to investigate whether effective HDAC3 inhibitors can improve the anti-tumor activities of AIs in endocrine-resistant breast cancer patients with upregulated ANKRD11 expression.

## Methods

**Cell culture.** Human ER+ breast cancer cell lines MCF-7 and T47D were obtained from the American Type Culture Collection (Manassas, VA, USA). Cell lines were authenticated using short tandem repeats analysis at passage 20. And cells at passage 25–31 were used in this study. Negative mycoplasma contamination of the cells was confirmed using TaKaRa PCR Mycoplasma Detection Set (TaKaRa, USA) at every-three passages. PCR Mycoplasma Detection Set (cat. no. 6601; Takara Bio, Inc.). MCF-7 and T47D were cultured in DMEM medium (Life Technologies, USA) supplemented with 10% fetal bovine serum and 1% penicillin-streptomycin mixture (Invitrogen, Carlsbad, CA), and maintained in a humidified atmosphere containing 5% $CO_2$ at 37 °C.

**Establishment of LTED cell lines.** Cells were plated in 6-well plate at a density of $2 \times 10^5$ per well and culture for 24 h (hrs). To mimic the resistance to aromatase inhibitor, MCF-7 and T47D cell lines were cultured in phenol red free RPMI-1640 containing 10% charcoal-stripped steroid-depleted fetal bovine serum and 1% penicillin-streptomycin mixture for over a year. When cell growth was no longer inhibited by estrogen deprivation, the LTED cell lines was considered successfully established to mimic AI-resistance. Then they were maintained in same estrogen-deprivation medium for routine culture. MCF-7 TAMR cell line was established as previously reported. Briefly, $2 \times 10^5$ cells cultured in 6-well plate were washed with PBS and cultured with RPMI-1640 containing 0.1 μM 4-OH tamoxifen. Finally, when the growth of the cells cannot be inhibited with 4-OH tamoxifen which was gradually increased up to 1 μM, the TAMR cell line was established[53,57].

**Antibodies and biochemical products.** The following primary antibodies were used: anti-SERPINA3 (Cat. ET1705-47, Huabio, China, 1:1000), anti-ANKRD11

(CSB-PA757769LA01HU, Cusabio, China, 1:1000), anti-HDAC3 (ET1610-5, Huabio, China, 1:1000), anti-acetylated H3K9 (AG3948, Beyotime, China, 1:1000), anti-histone H3 (9712, Cell Signaling, USA, 1:1000), anti ERα (8644T, Cell Signaling, USA, 1:1000). RGFP966 was purchased from Macklin (R883017, Macklin, China). Letrozole, exemestane and anastrozole were purchased from Selleck (S1235, S1196, S1188, Selleck Chemicals, China).

**Small interfering RNA (siRNA) studies**. Cells were plated in 6-well or 12-well plate at densities of ~$2 \times 10^5$ or $1 \times 10^5$ per well. 24 h later, cells were transfected with 40 nM siRNA using Lipofectamine® RNAiMAX Transfection Reagent (Life Technologies, USA) in complete medium. Medium was changed after 24 h and every other day thereafter until the end of the experiment. Non-targeting siRNA (5′-UUCUCCGAACGUGUCACGU-3′) was used as a nonsense control. The sequences of siRNAs targeting specific genes are listed in Table 3:

**Western blot analysis**. Cells were lysed in RIPA lysis buffer (Beotime, China) supplemented with protease and phosphatase inhibitors (Life Technologies, USA). Protein samples were separated using polyacrylamide gels and transferred to polyvinylidene fluoride membranes. After being blocked with 5% (weight/volume) non-fat dry milk, the membranes were incubated with primary antibodies, washed, and further incubated with secondary antibodies conjugated with horseradish peroxidase. Protein bands were detected by chemiluminescence and their intensities were determined by ImageJ software. GAPDH was used as the loading control.

**Quantitative RT-PCR (QPCR)**. RNA was extracted by the TRIZOL-chloroform extraction method from cells and used as templates for cDNA synthesis using the first-stranded cDNA synthesis kit (Invitrogen, Carlsbad, CA, USA). Real-time PCR was performed in a LightCycler480 System using a SYBR Premix ExTaq kit (Takara, Shiga, Japan). GAPDH was used as an internal control. Gene-specific primers were purchased from IGE Biotechnology (Guangzhou, China) and listed in Table 4:

**Cell counting**. Cells grown in 12-well plate were harvested and 20 μL of the mixture was added into one chamber of the five-chambered Countstar slide (Countstar, Shanghai, China). The number of cells was measured with automated cell counter (Countstar, Shanghai, China).

**Plate colony formation assay**. Cells were seeded into a 6-well plate at a density of $1 \times 10^3$ to $1 \times 10^4$ cells per well and treated with tamoxifen or estrogen-free medium for 7–14 days after transfection or not. Then medium was removed and cells were fixed with 4% paraformaldehyde for 10 min and stained with 0.5% crystal violet solution for 10 min. Images were obtained from a scanner.

**Construction of plasmids and luciferase activity assay**. ERα over-expressing plasmids pCDH-CMV-MCS-EF1-PURO-ESR1 and its empty vector pCDH were purchased from OriGene (Rockville, MD). The 1-kb fragment upstream of SERPINA3 transcription start point and its corresponding mutant type in which the predicted ERα binding sequence GAGGTTGCAGTGAGCTG was mutated into TTATATTTTTATTGTAA were synthesized and cloned into the pGL3-basic vector respectively (IGE Biotechnology, Guangzhou, China). SERPINA3 over-expressing plasmid was constructed by IGE Biotechnology with using pCDH as vector backbone. Dual luciferase reporter assay was performed by co-transfection of pGL3-promoter plasmids, ERα over-expressing plasmids or its empty vector, and pRL vectors using Dual-luciferase Reporter Assay System (Promega, Madison, WI) according to manufacturer's instructions. E2 (100 nM) was simultaneously added into the medium. Co-transfection with pCDH empty vector was taken as negative control.

**Co-immunoprecipitation (Co-IP) analysis**. Per 1 mg cell lysates were precleared by shaking incubating with 20 μl of 1:1 slurry of protein A/G agarose (Santa Cruz) for 2 h at 4 °C. Precleared samples were then incubated with primary antibodies or normal mouse IgG (control) overnight, followed by pulling-down with 40 μl protein A/G agarose beads for an additional 4 h. The beads were then collected by centrifugation, washed four times with lysis buffer and eluted in 2× sample laemmli buffer prior to western blot analysis.

**Statistics and reproducibility**. Downloaded expression microarray data of clinical samples from Gene Expression Omnibus database were quantile normalized and log2 transformed using R project. Subsequent graphs and differences analyses among groups were performed using Graphpad Prism v9.3.0 with paired Student's $t$ test. Kaplan–Meier survival curves were graphed using Sangerbox online tool, differences were examined using log-rank test. Western blot signal was quantified by image J. All data are representative of $n = 3$ biologically independent experiments and presented as mean ± SD in Origin 2019. Differences between samples were evaluated by Student's $t$ test (two groups) using Graphpad Prism v9.3.0, a two-tailed $p$-value < 0.05 was considered statistically significant.

**Reporting summary**. Further information on research design is available in the Nature Portfolio Reporting Summary linked to this article.

## Data availability
Gene array analyses identified 366 differentially expressed genes (DEGs) between MCF-7/S0.5 and its TAMR cell lines were obtained from Elias D's study[37]. DEGs between luminal breast cancer xenografts HBCx22, HBCx34 and their TAMR or ovariectomy resistant derivatives were obtained from Cottu P's study[38]. To identify the DEGs in human ER+ breast cancer before and after pre-surgical AI or tamoxifen treatment, the expression profiling data of GSE105777, GSE15347 and GSE147271 were obtained from the Gene Expression Omnibus database. Raw expression values provided by the datasets were all quantile normalized and log2 transformed for subsequent analysis. To screen the genes indicating poor prognosis in ER+ breast cancer, expression profiling dada of 4861 patients from different studies were included for prognostic analysis by using GenExMiner v4.9 database. Detailed information on the different patient populations could be found in Supplementary Data 2. Microarray data of 1057 breast tumors, which was retrieved from TCGA database, were used to evaluate prognostic value of SERPINA3 and ANKRD11 on overall survival in different subtypes of breast cancer, including luminal A, luminal B and ER−, by using a Kaplan–Meier survival plot. The baseline characteristics of the 1057-patients cohort were shown in Supplementary Data 1. Gene expression correlations were analyzed by using publicly available online databases Molecular Taxonomy of Breast Cancer International Consortium (METABRIC) in cBioPortal. Total 10,030 patients' microarray data of different origins was used to perform expression analysis of SERPINA3 and ANKRD11 in ER+ and ER− breast cancer in GenExMiner v4.9 database. Detailed information on the patient populations could be found in Supplementary Data 2. Microarray data generated in this study were deposited in GEO database (GSE234546). Uncropped western blot images are available in Supplementary Figs. 6, 7. The numerical data (source data) that makes up the bar graphs in the paper is organized in Supplementary Data 3. The newly generated SERPINA3 overexpressing plasmid pCDH-SERPINA3 in this study was constructed and preserved by IGE Biotechnology (Guangzhou, China).

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

## Acknowledgements

This work was supported by the National Natural Science Foundation of China (82230057, 82061148016, 82272859, 81872141), and Beijing Medical Award Foundation (YXJL-2020-0941-0760).

## Author contributions

J.Z. and M.Z. conceived of the study, carried out major part of experiments, and drafted the manuscript; Q.W., Y.Y.D. and N.L. participated in part of data acquisition; Y.L. coordinated the research activity and revised the manuscript; Q.L. supervised the research, revised the manuscript, and acquired funding support for this study. All authors read and approved the final manuscript.

## Competing interests

The authors declare no competing interests.
