## [Peer review file · Communications Biology]

Reviewers' comments:

Reviewer #1 (Remarks to the Author):

Zhou et al. found that SERPINA3 and its target gene ANKRD11 contributed to aromatase inhibitor resistance. The finding is interesting, however, the data in this manuscript cannot fully support the conclusions. Some issues need to be addressed.

1. Why are LTED cells considered as aromatase inhibitors (AI)-resistance cells?

2. Although the level of SERPINA3 was decreased in BC samples after treatment with AI (Fig 2F), and "lower expression of SERPINA3 was specifically related with poor prognosis of ER+ BC (Fig 2G), the data did not support the conclusion that "SERPINA3 is negatively associated with endocrine sensitivity".

3. Fig 3, authors demonstrated that SERPINA3 was regulated by estrogen receptor α . Is there any change in the level of estrogen receptor α in endocrine resistance cells? The data should be provided.

4. Compared with parent cells, how about the change of SERPINA3 and AKRD11 in LTED and TAMR cells?

5. The efficacy of tamoxifen, aromatase inhibitors or E2 should be observed in AKRD11 overexpressed cells.

6. Knock down AKRD11 in SERPRINAS silencing MCF-7 and T47D cells, and then evaluate the efficacy of E2, tamoxifen, or aromatase inhibitors in these cells.

7. Fig 5f, the effect of "E2 + RGFR966" on proliferation also should be evaluated in control cells and SERPINA3-knockdown cells.

Reviewer #2 (Remarks to the Author):

In this work, the authors show that loss of SERPINA3 expression leads to aromatase inhibitor (AI) resistance, and that this resistance is facilitated by ANKRD11 which activates HDAC3 (and hence loss of H3K9Ac). The work is highly relevant and a valuable contribution to the knowledge of AI resistance, and important for future studies of AI resistance in BC. The manuscript is also well written and the logic is easy to follow. Some concerns must be addressed before publication.

Major (more like moderate) concerns

In other works (e.g. <https://pubmed.ncbi.nlm.nih.gov/25625755/> and others) AI resistance is induced by actual treatment with AIs. Since the target of AIs in postmenopausal women is mainly the adipose tissue and the effect is decreased estrogen concentration in the blood, the strategy applied in the present work makes a lot of sense. However, since this is not established knowledge in the field, the authors should expand their discussion on how estrogen deprivation is representative for AI resistance (and how it may not be). This should be done both upfront (in the introduction) and in the discussion, and comparisons should be made to alternative methods.

The analyses made on patient material are not described in the Materials and Methods. All these analyses must be properly explained. The PDXs are not described, either. The data and analyses used in Figure 2F and 2G should be properly explained and cited in M&M. Even if the data in Figure 2G is used through the bc-GenExMiner, the patient cohort in the analysis should be described and cited. A short summary of patient characteristics in Figure 2F should also be included.

In Figure 2C, data should also be shown for MCF7.

The analysis in Figure 2G for ER positive should also be further stratified into Luminal A and Luminal B, since proliferation could be a confounding factor for the survival analysis. This data should be presented in the main text, while the ER negative could be moved to supplementary if needed. Also for figure 4J, the ER negative should be moved to supplementary, and analyses for Luminal A and Luminal B separately should be included.

When investigating how SERPINA3 causes AI resistance, why do the authors focus only on genes negatively correlated to SERPINA3 expression?

Minor:

Figure 2D should be transposed (rows and columns), to match fig 2C

Figure 3 legend: I) should be H)

In the sentence "The data from GSE147271 showed a remarkable downregulation of SERPINA3 and 278 up-regulation of ANKRD11 after endocrine treatment (Fig. 4I)" I would remove "remarkable". It's non-quantitative, and quite subjective.

Reviewer #3 (Remarks to the Author):

This paper aimed to demonstrate SERPINA3-ANKRD11-HDAC3 pathway involved in aromatase inhibitor resistance and reversed in HDAC3 inhibition. The experiments were well-organized in two ER-positive cell lines. However, there are concerns about the cell lines used in the study, which are the main models to exhibit the resistant mechanism in the study. Thus, please provide more details on how the author characterized LTED cell lines and how to demonstrate that these cell lines can mimic AI-resistance.

We sincerely thank you for giving us the opportunity to revise the manuscript, and we deeply appreciate the insightful and constructive comments from the reviewers. In this updated manuscript, we have thoroughly revised the manuscript according to the reviewers' comments. Please see below our point-by-point responses.

Author's reply to the Reviewers

RE: Manuscript ID **COMMSBIO-22-4106-T**

Title: SERPINA3-ANKRD11-HDAC3 pathway contributes to aromatase inhibitor resistance in breast cancer that could be reversed by HDAC3 inhibition

Authors: Jing Zhou, Mengdi Zhu, Qi Wang, Yiyuan Deng, Nianqiu Liu, Yujie Liu and Qiang Liu

Referee Comments	Authors' Reply
Reviewer #1: Zhou et al. found that SERPINA3 and its target gene ANKRD11 contributed to aromatase inhibitor resistance. The finding is interesting, however, the data in this manuscript cannot fully support the conclusions. Some issues need to be addressed:	We sincerely thank the reviewer for these valuable comments.
1. Why are LTED cells considered as aromatase inhibitors (AI)-resistance cells?	In clinical practice, aromatase inhibitors (AIs) are typically administered in postmenopausal breast cancer patients to inhibit estrogen synthesis in the peripheral tissues such as adipose tissue, breasts and skin, thereby decrease blood estrogen concentrations and prevent cancer cell growth (Ma CX, et al. Nat Rev Cancer. 2015), but not directly act on the cancer cells themselves because of low or no expression of endogenous aromatase in cancer cells (Chen X, et al. Nat Commun. 2022), which could be manifested by the insensitivity of ER+ breast cancer cells to therapeutic blood concentrations of the letrozole (50-100 ng/ml), anastrozole (10-40 ng/ml) and exemestane (10-20 ng/ml) (Awada A, et al. Eur J Cancer. 2008; Valle M, et al. Br J Clin Pharmacol. 2005; Gervasini G, et al. Br J Clin Pharmacol. 2017) (Supplementary Fig. S3). Long-term-

	estrogen-deprivation (LTED) model was established by maintaining ER-positive breast cancer cells in estrogen deprivation condition for over one year to more closely mimic the physiological estrogen-depletion situation induced by AI therapy in patients than other models. The validity of LTED model in mimicking AI-resistance has been widely accepted, as evidenced by the use of this model in a number of high-quality AI resistance studies (Chen X, et al. Nat Commun. 2022; Williams MM, et al. Cell Death Dis. 2018; Du T, et al. Breast Cancer Res. 2018; Bhola NE, et al. Cancer Res. 2015; Fox EM, et al. Cancer Res. 2011; Miller TW, et al. J Clin Invest. 2010). In another way of establishing AI-resistant model, specific AI was used to directly treat ER-positive breast cancer cells with ectopically overexpressed aromatase. This methodology is artificial and very different from factual physiological environment. The exclusive resistance to each specific AI of letrozole-, exemestane- and anastrozole-resistant MCF-7 cell lines (Masri S et al. Cancer Res. 2008) revealed the limitations of this model in mimicking patients' AI-resistance because patients who are resistant to letrozole are also resistant to anastrozole. This issue has now been explained in the introduction (Page 6, lines 89-92) and discussion (Pages 18-19, lines 364-384) part in the revised manuscript.
2. Although the level of SERPINA3 was decreased in BC samples after treatment with AI (Fig 2F), and “lower expression of SERPINA3 was specifically related with poor prognosis of ER+ BC (Fig 2G), the data did not support the conclusion that “SERPINA3 is negatively associated with endocrine sensitivity”.	We thank the reviewer for pointing out this issue. In light of this advice, we changed the original sentence “Furthermore, lower expression of SERPINA3 is specifically related with poor prognosis of ER+ BC, but not ER negative (ER-) subtype according to the bc-GenExMiner v4.8 database (Fig. 2g), suggesting SERPINA3 expression is negatively associated with endocrine sensitivity in ER+ BC.” to the revised sentence “Furthermore, according to the data retrieved from TCGA database (Supplementary Table. S2), lower expression of SERPINA3 is specifically correlated with poorer prognosis in both luminal A and luminal B subtypes of ER+ breast cancer (Fig. 2g), but not in ER- subtype (Supplementary Fig. S1). All

	above results together suggest that the expression of SERPINA3 is negatively associated with endocrine resistance in ER+ BC.” As demonstrated in the revised sentence, we want to express that all results in Fig.2, including that SERPINA3 knockdown endowed breast cancer cells with estrogen deprivation resistance (Fig. 2b and c), while SERPINA3 overexpression restored endocrine sensitivity in LTED cells (Fig. 2d), and the observed diminished expression of SERPINA3 in LTED cells compared with their parent counterparts (Fig. 2e), as well as Fig. 2f and g mentioned in the question, jointly indicate that “SERPINA3 is negatively associated with endocrine resistance”. Please refer to the revised manuscript Page 14, lines 268-273. TCGA database was used in the revised manuscript because it has more cases that show similar findings and allows separate analysis of luminal A and luminal B subtypes.
3. Fig 3, authors demonstrated that SERPINA3 was regulated by estrogen receptor α. Is there any change in the level of estrogen receptor α in endocrine resistance cells? The data should be provided.	We agree that estrogen deprivation did not necessarily suppress estrogen receptor α (ERα) in endocrine resistant cells, given that overexpression/overactivation of ERα is a common strategy in developing endocrine resistance. Variable changes of ER levels in different breast cancer LTED cell lines had been reported (Miller TW, et al. J Clin Invest. 2010). Thus, it is necessary to examine expression changes of ERα in endocrine resistant cells before attributing the diminished SERPINA3 expression to ERα pathway inhibition. As recommended, QPCR and western blot were performed to detect the expression level of ERα in LTED and their parental cells. The results from both experiments showed that LTED cell lines possessed decreased ERα expression levels when compared with their parent MCF-7 or T47D cells. The results were presented in Fig. 3c and d and described in the revised manuscript as “In addition, the expression of SERPINA3 was downregulated in LTED cells (Fig. 1d and 2e) in which ERα activity was inhibited due to the decreased ERα expression (Fig. 3c and d) and estrogen deprivation, implying an inhibition effect of ERα pathway on SERPINA3 expression.” (Pages 14-15, Lines 283-

286). This finding, together with other results in Fig. 3, support our conclusion that SERPINA3 expression was inhibited by ER α pathway.

Fig. 3 Supplemented (c) QPCR and (d) western blot results showed that ER α expression was downregulated in endocrine-resistant LTED cells as compared with their parental cells.

4. Compared with parent cells, how about the change of SERPINA3 and AKRD11 in LTED and TAMR cells?

We want to thank the reviewer for raising this question. The expression changes of SERPINA3 and AKRD11 in LTED cells have been shown in Fig. 2e and Fig. 4g respectively. LTED cells exhibited inhibited expression of SERPINA3 and enhanced expression of ANKRD11 when compared with their parent cells. For TAMR cells, as this project mainly focused on AI resistance study, and to avoid duplication with another ongoing research work in our group, tamoxifen resistance will not be particularly discussed in this study.

5. The efficacy of tamoxifen, aromatase inhibitors or E2 should be observed in AKRD11 overexpressed cells.

To observe the gain-of-function effects of ANKRD11, we had tried to construct ANKRD11-overexpressing plasmids several times previously. However, all attempts had failed because the coding sequence of ANKRD11 is too long (7992 bp) to be amplified by PCR (Ramos A L, et al. *Journal of Biological Research-Bollettino della Società Italiana di Biologia Sperimentale*. 2018) and cloned into expression vectors (Konermann S, et al. *Nature*. 2015). In addition, it can also be an intractable problem to transform such a large plasmid into competent cells (Li M, et al. *Microb Cell Fact*. 2012). Therefore, we are sorry that only loss-of-function studies were used to demonstrate the positive role of ANKRD11 in estrogen-deprivation-resistance development.

6. Knock down AKRD11 in SERPRINAS

We appreciate this constructive suggestion. As suggested, to observe

silencing MCF-7 and T47D cells, and then evaluate the efficacy of E2, tamoxifen, or aromatase inhibitors in these cells.

if ANKRD11 knock-down could abolish SERPINA3 inhibition induced estrogen-deprivation-resistance, ANKRD11 was simultaneously knocked down with SERPINA3 or not in MCF-7 and T47D respectively. After 5-day estrogen deprivation treatment or not, cell proliferation was evaluated by cell counting. The results showed that, as supplemented Fig. 4d, ANKRD11 inhibition restored sensitivity to estrogen deprivation in SERPINA3 knock-down breast cancer cells. Manuscript was modified accordingly as “*Simultaneous knock-down of ANKRD11 with SERPINA3 significantly abolished SERPINA3 inhibition induced E2-deprivation resistance in MCF-7 and T47D cells (Fig. 4d).*” (Page 16, Lines 315-317).

Fig. 4 (d) A supplementary experiment in which ANKRD11 were simultaneously knock-down with SERPINA3 showed that ANKRD11 suppression significantly abolished SERPINA3 inhibition induced E2-deprivation resistance in MCF-7 and T47D cells.

7. Fig 5f, the effect of “E2 + RGFR966” on proliferation also should be evaluated in control cells and SERPINA3-knockdown cells.

As recommended, the experiment was re-designed to include another E2+RGFP966 treatment group. The result was shown in revised Fig. 5f.

Fig. 5 (f) The experiment was re-designed to include E2+RGFP966 treatment group. The results showed that RGFP966 significantly inhibited ER+ BC cell growth if SERPINA3 was knocked down, with or without E2 incubation. And this experiment was repeated in T47D cells.

Reviewer #2: In this work, the authors show that loss of SERPINA3 expression leads to

We would like to express our sincere thanks for the reviewer’s positive

aromatase inhibitor (AI) resistance, and that this resistance is facilitated by ANKRD11 which activates HDAC3 (and hence loss of H3K9Ac). The work is highly relevant and a valuable contribution to the knowledge of AI resistance, and important for future studies of AI resistance in BC. The manuscript is also well written and the logic is easy to follow. Some concerns must be addressed before publication.	comments on the manuscript and constructive suggestions.
Major (more like moderate) concerns:	
1. In other works (e.g. https://pubmed.ncbi.nlm.nih.gov/25625755/ and others) AI resistance is induced by actual treatment with AIs. Since the target of AIs in postmenopausal women is mainly the adipose tissue and the effect is decreased estrogen concentration in the blood, the strategy applied in the present work makes a lot of sense. However, since this is not established knowledge in the field, the authors should expand their discussion on how estrogen deprivation is representative for AI resistance (and how it may not be). This should be done both upfront (in the introduction) and in the discussion, and comparisons should be made to alternative methods.	Please see the detailed response to the concerns regarding long-term-estrogen-deprivation (LTED) model in the answers to the point 1 from Reviewer #1.
2. The analyses made on patient material are not described in the Materials and Methods.	We sincerely apologize for the negligence and thank the reviewer for pointing out. Revisions have been made accordingly. A “Data

All these analyses must be properly explained. The PDXs are not described, either. The data and analyses used in Figure 2F and 2G should be properly explained and cited in M&M. Even if the data in Figure 2G is used through the bc-GenExMiner, the patient cohort in the analysis should be described and cited. A short summary of patient characteristics in Figure 2F should also be included.	retrieval” section has been added in the Materials and Methods in the revised manuscript (Pages 7-8, lines 114-135). And “Statistics and reproducibility” section in the Materials and Methods was also introduced in more detail (Pages 11-12, lines 219-227). The references that provide the original expression profiling data were accordingly cited (Reference 40-42). Brief characteristics description on patients in GSE105777, GSE153470 and GSE147271 was also included in figure legends.
3. In Figure 2C, data should also be shown for MCF7.	As recommended, the effect of SERPINA3 knock-down on estrogen deprivation resistance was examined in MCF-7 using colony formation assay, as such in T47D. The result was shown in Fig. 2c. Manuscript was modified accordingly at Page 13, lines 260-262: Plate colony formation assay performed in MCF-7 and T47D cells confirmed that knock-down of SERPINA3 expression would result in E2-deprivation-resistance (Fig. 2c).   Fig. 2 (c) Plate colony formation assay after SERPINA3 knockdown which performed in T47D cells was repeated in MCF-7 cells, and similar results from two different cell lines were shown.
4. The analysis in Figure 2G for ER positive should also be further stratified into Luminal A and Luminal B, since proliferation could be confounding factor for the survival analysis. This data should be presented in the main text, while the ER negative could be moved to supplementary if needed. Also for	We appreciate this valuable comment. As suggested, SERPINA3 and ANKRD11 prognosis analysis in ER positive breast cancer were further stratified into luminal A and luminal B (Fig. 2g and 4k) by using expression profiling data of 1,057 breast tumors retrieved from TCGA database. The baseline characteristics of the 1057 breast cancer patients was shown in Supplementary Table S2. Prognosis analysis of these two gene in ER negative breast cancer were moved to

figure 4J, the ER negative should be moved to supplementary, and analyses for Luminal A and Luminal B separately should be included.

Supplementary Fig. S1 and S2 as suggested. Manuscript was revised accordingly at Page 14, lines 268-271; Pages 16-17, lines 328-331; Page 20, lines 413-418.

Fig. 2 (g) The prognostic role of SERPINA3 expression was separately evaluated in luminal A and luminal B BC, but not in all ER+ BCs. Results showed that lower expression of SERPINA3 is correlated with poorer prognosis in both luminal A and luminal B subtypes of ER+BC, but not in ER- BC (shown in supplementary Fig. S1).

Fig. 4 (k) The prognostic role of ANKRD11 expression was separately evaluated in luminal A and luminal B BC, but not in all ER+ BCs. Results showed that higher expression of ANKRD11 is correlated with poorer prognosis in luminal A BC but opposite outcome in luminal B and ER- BC (shown in supplementary Fig. S2).

5. When investigating how SERPINA3 causes AI resistance, why do the authors focus only on genes negatively correlated to SERPINA3 expression?

We thank the reviewer for this thoughtful question. According to cBioPortal database, there are 61 and 293 genes are respectively negatively and positively correlated with SERPINA3 expression in breast cancer. We think a considerable number of the positively correlated genes may be responsive genes of ER α , but not actual SERPINA3 regulated genes. Thus, we focused on the 61 genes that are negatively related with SERPINA3 and upregulated in AI-resistant cells. And we found ANKRD11 was involved in estrogen-deprivation-resistance development. However, we cannot rule out the possibility that the endocrine resistance could be regulated by the genes which positively correlated with SERPINA3 expression and this worth of further study.

Minor:	First we would like to express our sincere gratitude for the reviewer's careful and thorough review on this manuscript. We appreciate these corrections very much.																																																
6. Figure 2D should be transposed (rows and columns), to match fig 2C.	Fig.2d has been transposed accordingly. Please refer to Fig. 2d in the revised manuscript.  c     MCF-7 T47D    Si Ctrl SERPINA3 Si #1 SERPINA3 Si #2 Si Ctrl SERPINA3 Si #1 SERPINA3 Si #2     +E2         -E2          d     MCF-7 LTED T47D LTED    pCDH pCDH-SERPINA3 pCDH pCDH-SERPINA3     +E2       -E2         Fig. 2 The alignment of the figures in Fig. 2d has been transposed to match Fig.2c as suggested.		MCF-7			T47D				Si Ctrl	SERPINA3 Si #1	SERPINA3 Si #2	Si Ctrl	SERPINA3 Si #1	SERPINA3 Si #2	+E2							-E2								MCF-7 LTED		T47D LTED			pCDH	pCDH-SERPINA3	pCDH	pCDH-SERPINA3	+E2					-E2				
	MCF-7			T47D																																													
	Si Ctrl	SERPINA3 Si #1	SERPINA3 Si #2	Si Ctrl	SERPINA3 Si #1	SERPINA3 Si #2																																											
+E2																																																	
-E2																																																	
	MCF-7 LTED		T47D LTED																																														
	pCDH	pCDH-SERPINA3	pCDH	pCDH-SERPINA3																																													
+E2																																																	
-E2																																																	
7. Figure 3 legend: I) should be H).	This error has been corrected in the revised manuscript.																																																
8. In the sentence "The data from GSE147271 showed a remarkable downregulation of SERPINA3 and 278 up-regulation of ANKRD11 after endocrine treatment (Fig. 4I)" I would remove "remarkable". It's non-quantitative, and quite subjective.	The word "remarkable" has been removed from this sentence as suggested, please refer to Page 16, lines 325-327: "The data from GSE147271 showed a downregulation of SERPINA3 and up-regulation of ANKRD11 after endocrine treatment (Fig. 4j)"																																																
Reviewer #3: This paper aimed to demonstrate SERPINA3-ANKRD11-HDAC3 pathway involved in aromatase inhibitor resistance and reversed in HDAC3 inhibition. The experiments were well-organized in two ER-positive cell lines.	We appreciate the reviewer's concern about the rationality of LTED model in mimicking AI-resistance. Please see the detailed response to the concerns regarding long-term-estrogen-deprivation (LTED) model in the answers to the point 1 from Reviewer #1. And as suggested, LTED cells were further characterized using colony formation assay and cell counting assay to examine the effects of E2 deprivation																																																

However, there are concerns about the cell lines used in the study, which are the main models to exhibit the resistant mechanism in the study. Thus, please provide more details on how the author characterized LTED cell lines and how to demonstrate that these cell lines can mimic AI-resistance.

(Fig.1a and b) and tamoxifen (TAM) treatment (Fig.1c and d) on their growth and survival. Please refer to Pages 12-13, lines 237-243 in revised manuscript for detailed description. In addition, the expression of ER α was shown to be downregulated in both LTED cells by QPCR (Fig.3c) and western blot (Fig. 3d) when compared with their parent cells.

Fig. 1 Characterization of LTED cells. Colony formation assay (a and c) and cell counting assay (b and d) showed that LTED cells of MCF-7 and T47D are resistant to E2 deprivation (a and b) and tamoxifen (TAM) treatment (c and d).

Fig. 3 Supplemented (c) QPCR and (d) western blot experiments showed that ER α was downregulated in endocrine-resistant LTED cells as compared with their parental cells.

REVIEWERS' COMMENTS:

Reviewer #1 (Remarks to the Author):

The authors were well addressed reviewers' concern.

Reviewer #2 (Remarks to the Author):

I thank the authors for revising the manuscript. I have no further concerns.

Reviewer #3 (Remarks to the Author):

The authors addressed that the LTED model resembled the clinical setting of patients who received AI therapy and caused estrogen deprivation. The authors also mentioned the experiments on colony formation assays that the cells were resistant to estrogen deprivation and tamoxifen together with QPCR of ESR1 to assume that the cells were resistant to AI. How do the authors confirm that this result is AI resistance, not tamoxifen resistance?

We are very pleased to hear that our manuscript (ID COMMSBIO-22-4106A) is acceptable for publication in *Communications Biology* with minor revision. And we appreciate the comments from the reviewer #3. In this updated manuscript, we have revised the manuscript according to the reviewer's comments and format requirements. Please see below our response to the reviewer #3.

Author's reply to the Reviewers

RE: Manuscript ID **COMMSBIO-22-4106A**

Title: SERPINA3-ANKRD11-HDAC3 pathway induced aromatase inhibitor resistance in breast cancer can be reversed by HDAC3 inhibition

Authors: Jing Zhou, Mengdi Zhu, Qi Wang, Yiyuan Deng, Nianqiu Liu, Yujie Liu and Qiang Liu

Referee Comments	Authors' Reply
Reviewer #3: The authors addressed that the LTED model resembled the clinical setting of patients who received AI therapy and caused estrogen deprivation. The authors also mentioned the experiments on colony formation assays that the cells were resistant to estrogen deprivation and tamoxifen together with QPCR of ESR1 to assume that the cells were resistant to AI. How do the authors confirm that this result is AI resistance, not tamoxifen resistance?	We sincerely thank the reviewer for doing an in-depth reading of our work and providing this valuable comment. And we hope the answer below could meet the reviewer's concerns. The LTED model, which has been proven to resistant to low plasma E2 concentration in vivo [Shim WS, et al. Endocrinology. 2000], was established by long-term estrogen deprivation. Therefore, the cells developed in this process are results of estrogen deficiency-resistance (or AI resistance) screening, and naturally are also resistant to tamoxifen that works by blocking E2 signaling. Two other reports showed similar findings that tamoxifen resistance was also identified as a property of LTED cells [Martin LA, et al. Endocr Relat Cancer. 2005; Henriques Palma GB & Kaur M. Anticancer Res. 2022]. However, to avoid ambiguity or misinterpretation that may arise in the tamoxifen resistance of AI-resistant LTED cells, we removed the tamoxifen part from Figure 1.